# Delayed Momentum Aggregation: Communication-efficient Byzantine-robust Federated Learning with Partial Participation

## Abstract

Federated Learning (FL) allows distributed model training across multiple clients while preserving data privacy, but it remains vulnerable to Byzantine clients that exhibit malicious behavior. While existing Byzantine-robust FL methods provide strong convergence guarantees (e.g., to a stationary point in expectation) under Byzantine attacks, they typically assume full client participation, which is unrealistic due to communication constraints and client availability. Under partial participation, existing methods fail immediately after the sampled clients contain a Byzantine majority, creating a fundamental challenge for sparse communication. First, we introduce *delayed momentum aggregation*, a novel principle where the server aggregates the most recently received gradients from non-participating clients alongside fresh momentum from active clients. Our optimizer D-Byz-SGDM (Delayed Byzantine-robust SGD with Momentum) implements this delayed momentum aggregation principle for Byzantine-robust FL with partial participation. Remarkably, experiments on deep learning tasks showed our method not only maintained stable convergence under various Byzantine attacks, but also outperformed standard FL methods with partial participation in non-Byzantine settings.

## 1 Introduction

Federated Learning (FL) enables collaborative training across many clients without centralizing raw data, and has become a standard approach when privacy, bandwidth, or governance constraints prevent data pooling (Kairouz et al., 2021; McMahan et al., 2017). Its central idea is to transmit gradients rather than raw data. Specifically, each client computes the gradient using their local dataset and sends it to the central server. Then, the central server computes the average of the gradients and updates the parameters. Since its proposal, FL has attracted many optimization researchers and has been widely studied in areas such as communication compression (Mishchenko et al., 2024; Khirirat et al., 2018; Horváth et al., 2023; Stich et al., 2018; Alistarh et al., 2017; Albasyoni et al., 2020; Li et al., 2021; Fatkhullin et al., 2023), data heterogeneity (Karimireddy et al., 2020b; Pu & Nedić, 2021; Takezawa et al., 2022; Cheng et al., 2024; Li et al., 2020; Yang et al., 2021; Wang et al., 2020; Zhang et al., 2021; Haddadpour et al., 2021; Alghunaim, 2024), accelerated methods (Kovalev et al., 2022; Jiang et al., 2024; d'Aspremont et al., 2021; Güler, 1992; Nesterov, 2018; Lin et al., 2015; Monteiro & Svaiter, 2013), and Byzantine-robust FL, including defenses for homogeneous data (Blanchard et al., 2017a; Mhamdi et al., 2018; Damaskinos et al., 2019; Yin et al., 2018; Pillutla et al., 2022; Bernstein et al., 2019; Alistarh et al., 2018; Mhamdi et al., 2021; Karimireddy et al., 2021) and heterogeneous data (Sattler et al., 2020; Xie et al., 2019b; Chen et al., 2018; Rajput et al., 2019; Data & Diggavi, 2021a;b; Li et al., 2019; Acharya et al., 2022; El-Mhamdi et al., 2021; Yang & Li, 2021; Allouah et al., 2023).

Due to the nature of FL, where a large number of clients participate in the training process, it is vulnerable to clients that behave incorrectly, commonly referred to as Byzantine clients (Kairouz et al., 2021; Lamport et al., 2019). For instance, some clients may be faulty, while others may act maliciously to disrupt training. Under Byzantine failures, naive averaging is notoriously brittle:

even a single Byzantine client can significantly skew the aggregated model updates. To address this issue, a large body of work has proposed Byzantine-robust FL methods (Blanchard et al., 2017a;b; Allouah et al., 2023; Karimireddy et al., 2021), which replace simple averaging with robust aggregation rules at the central server. A robust aggregator guarantees that, as long as the majority of inputs come from honest clients, the aggregation output remains close to the true average of the honest clients' parameters, regardless of the values sent by malicious clients. Thanks to these robust aggregation techniques, Byzantine-robust FL can maintain convergence guarantees, despite the presence of Byzantine clients.

However, most of these existing Byzantine-robust FL methods rely on the assumption that all clients participate in every round, which is unrealistic. Some clients may be temporarily unavailable, for example, due to unreliable connections or competing computational tasks (Kairouz et al., 2021; Bonawitz et al., 2017; Niu et al., 2020; Yan et al., 2024; Gu et al., 2021; Wang & Ji, 2022). Even if all clients were available, it is common practice to sample only a subset of the clients to reduce the communication overhead between the central server and the clients (Karimireddy et al., 2020b;a; Patel et al., 2022). When only a subset of clients participates, most existing Byzantine-robust FL methods fail to remain robust against Byzantine clients. Specifically, in the partial participation setting, the majority of the sampled clients can be malicious. In such a case, a robust aggregator may no longer provide a good estimation of the average of the honest clients' parameters. Only a few papers have studied Byzantine-robust FL with partial participation (Allouah et al., 2024; Malinovsky et al., 2024). Malinovsky et al. (2024) proposed a MARINA-style (SVRG/SAGA-family) optimizer with a specialized clipping strategy, showing tolerance even in rounds with a Byzantine majority. However, such MARINA/SVRG/SAGA or periodic full-gradient / large-minibatch schemes perform poorly for deep learning models (Defazio & Bottou, 2019). Allouah et al. (2024) proposed replacing the naive averaging in FedAvg (McMahan et al., 2017) with a Byzantine-robust aggregator. Their algorithm, however, relies on vanilla (non-momentum) SGD, which is vulnerable to time-coupled attacks (Baruch et al., 2019; Karimireddy et al., 2021), and it offers no mitigation when Byzantine clients form a majority.

In this paper, we tackle the challenge of Byzantine-robust FL with partial participation, aiming for a simple and practical solution. Our proposed method, D-Byz-SGDM (Delayed Byzantine-robust SGD with Momentum), is strikingly simple: at each aggregation step, the central server aggregates not only the gradients sent from the sampled clients but also the most recently received gradients from the non-sampled clients. As a result, this effectively aggregates the entire set of clients, thereby ensuring that the aggregation in which Byzantine clients constitute a majority never occurs during the training. Experiments on deep learning tasks show stable and robust training under both partial participation and Byzantine attacks.

We provide a comprehensive discussion of related work in Section 2 and proceed with the formal problem setup.

## 2 RELATED WORK

**Byzantine-robust FL under full participation.** Classical defenses replace naive averaging by robust aggregation rules such as Krum (Blanchard et al., 2017a), coordinate-wise median and trimmed-mean (Blanchard et al., 2017b), and geometric–median–based RFA (Pillutla et al., 2022); meta-rules like Bulyan further reduce adversarial leverage (Mhamdi et al., 2018). Yet these per-round defenses can be vulnerable to time-coupled attacks that inject small, undetectable biases which accumulate across rounds (Baruch et al., 2019; Xie et al., 2019a). A key development is to leverage history: Karimireddy et al. (2021) formalize such time-coupled failures and prove that momentum (together with robust aggregation) provably restores convergence; subsequent works refine the momentum view and resilient averaging (Farhadkhani et al., 2022). Heterogeneity (non-IID client data) exacerbates the problem: bucketing (Karimireddy et al., 2022) and nearest-neighbor mixing (NNM) (Allouah et al., 2023) are pre-aggregation mechanisms that systematically adapt IID-optimal rules (e.g., Krum, median, RFA) to the heterogeneous regime, closing gaps between achievable rates and lower bounds. Beyond aggregation, algorithmic alternatives include coding-theoretic redundancy (DRACO) (Chen et al., 2018) and filtering for non-convex objectives (Allen-Zhu et al., 2021; Alistarh et al., 2018). Complementing these meta-aggregation approaches that assume full participation, Dahan & Levy (2024b) propose an efficient *Centered Trimmed Meta-*

*Aggregator* (CTMA) that upgrades base robust aggregators to order-optimal performance at near-averaging cost, and couple it with a double-momentum estimator to establish theoretical guarantees within the stochastic convex optimization (SCO) framework for synchronous (full-participation) training.

**Partial participation, and local updates.** Partial participation makes robustness strictly harder because the sampled set occasionally contains a Byzantine majority. Early theory coupling Byzantine robustness with local steps shows that convergence can be ensured only when the sampled cohort has a sufficiently large honest fraction at each synchronization, e.g., $\varepsilon \leq 1/3$ corrupted among the $K$ active clients (Data & Diggavi, 2021b, Thm. 1), an assumption strained by client sampling. The interaction between client sampling, multiple local steps, and robust aggregation has since been analyzed in detail by Allouah et al. (2024), who quantifies how client sampling reshapes the effective number of Byzantine clients and shows regimes where standard robust aggregators suffice; however, these schemes omit momentum and do not mitigate time-coupled drift. Another concurrent line uses explicit MARINA/SVRG/SAGA periodic full-gradient/reference-gradient steps: by coupling robust aggregation with gradient-difference clipping and periodic full-gradient steps, Malinovsky et al. (2024) proves tolerance even when a sampled round is entirely Byzantine, at the cost of periodic heavier steps. From a statistical-efficiency angle, protocols with near-optimal rates under full participation have been derived via modern robust statistics (Zhu et al., 2023), and recent work explores communication compression jointly with robustness (Rammal et al., 2024; Gorbunov et al., 2023).

**Connection to MIFA.** MIFA (Gu et al., 2021) tackles arbitrary client unavailability by caching each client's latest update and substituting this surrogate when the client is absent, followed by naive averaging. Within our framework this is the instantiation where the robust aggregator is replaced by the mean and the client-side momentum weight is fixed at $\alpha = 1$, i.e., cached updates are used without attenuation. In our analysis this corresponds to a robustness constant $c = \infty$, making our generic upper bound vacuous and reflecting that naive averaging cannot deliver Byzantine guarantees. Moreover, MIFA omits momentum; even if paired with a robust aggregator, the lower bound of Karimireddy et al. (2021) would still apply, as momentum is necessary to overcome time-coupled Byzantine drift.

**Asynchrony, delayed gradients, and relevance to our staleness mechanism.** Analysis of asynchronous SGD (ASGD) formalizes *delayed/stale* gradients and shows that delays can be controlled via delay-aware stepsizes (Koloskova et al., 2022; Mishchenko et al., 2022). In the *Byzantine asynchronous* regime, recent work Dahan & Levy (2024a) develops a *weighted* robust-aggregation framework and, combined with a double-momentum estimator, proves optimal convergence in the smooth *convex homogeneous* (i.i.d.) setting (Dahan & Levy, 2024a). Importantly for assumptions, Dahan & Levy (2024a;b)'s analysis (both asynchronous and synchronous) operates over a *compact* feasible set (bounded diameter), which is stricter than the bounded-gradient conditions commonly adopted in FL theory.

Our setting is not asynchronous; nevertheless, partial participation induces *server-side staleness* because non-sampled clients contribute historical (per-client) gradients. This places our analysis close to the ASGD toolbox while tackling a distinct failure mode (occasional Byzantine-majority samples under subsampling) without trusted validation data. Technically, we leverage *per-client* stale gradients to preserve a history-coupled (global) momentum across rounds, complementing weighted robust aggregation in the asynchronous literature (Dahan & Levy, 2024a).

Relative to prior momentum-based defenses (Karimireddy et al., 2021; Farhadkhani et al., 2022) and heterogeneity fixes (Karimireddy et al., 2022; Allouah et al., 2023), we study the regime where clients refresh stochastically and Byzantine clients can transiently comprise the sampled majority. Compared to MARINA/SVRG/SAGA-style periodic full-gradient/reference-gradient approaches (Malinovsky et al., 2024), our method avoids periodic full-batch gradient computations, making it more practical and scalable for real federated learning deployments.

## 3 PRELIMINARY

**Notations.** Our notation largely follows (Koloskova et al., 2020; Karimireddy et al., 2022). We denote by $n$ the total number of clients, and for any positive integer $k$, let $[k] := \{1, 2, \ldots, k\}$. The set of good (non-Byzantine) clients is represented by $\mathcal{G} \subseteq [n]$ with cardinality $G := |\mathcal{G}|$. The Byzantine ratio is defined as $\delta := (n - G)/n$, and throughout this paper we assume $\delta < 1/2$. For each client $i$, let $\mathcal{D}_i$ denote the distribution of local data $\xi_i$ over parameter space $\Omega_i$. The local loss function is given by $f_i : \mathbb{R}^d \to \mathbb{R}$, defined as $f_i(x) := \mathbb{E}_{\xi_i}[F_i(x; \xi_i)]$ where $F_i : \mathbb{R}^d \times \Omega_i \to \mathbb{R}$ is the sample loss.

**Problem Definition.** We formalize the problem as follows:

$$\min_{x \in \mathbb{R}^d} \left\{ f(x) := \frac{1}{G} \sum_{i \in \mathcal{G}} f_i(x) \right\}$$

where $x \in \mathbb{R}^d$ denotes the model parameters and $\mathcal{D}_i$ represents the dataset distribution of client $i$. In general, $\mathcal{D}_i \neq \mathcal{D}_j$, reflecting data heterogeneity across clients.

**Byzantine-robust Learning under Full-Participation** The full participation setting serves as the theoretical foundation for Byzantine-robust federated learning, where the fundamental challenge is designing aggregation mechanisms that maintain convergence guarantees despite adversarial behavior. This setting provides clean theoretical analysis by eliminating client sampling complexities, establishing design principles for robust aggregation rules and performance benchmarks that inform practical algorithm design. The case of full client participation has been extensively studied in the literature (Karimireddy et al., 2022; Allouah et al., 2023; Gorbunov et al., 2023).

In this setting, robustness is typically achieved by replacing the simple average with a robust aggregation rule. While the precise definition of such aggregators may vary across works, we adopt the following notion from Karimireddy et al. (2022) and use it throughout this paper.

**Assumption 1** (($\delta, c$)-Robust Aggregator (Karimireddy et al., 2022; Malinovsky et al., 2024))**.** Let $\{X_1, X_2, \ldots, X_n\}$ be a set of random vectors. Suppose there exists a "good" subset $\mathcal{G} \subseteq [n]$ of size $G = |\mathcal{G}| > n/2$ such that

$$\mathbb{E}\|X_i - X_j\|^2 \leq \rho^2, \ \forall i, j \in \mathcal{G}.$$

Then the output $\hat{X}$ of a Byzantine-robust aggregator Agg satisfies

$$\mathbb{E}\| \mathrm{Agg}(X_1, \ldots, X_n) - \bar{X}\|^2 \leq c\delta\rho^2, \ \text{where} \ \bar{X} = \frac{1}{G} \sum_{i \in \mathcal{G}} X_i.$$

Importantly, this definition is not merely abstract. Karimireddy et al. (2022) prove (in Theorem 1) that well-known aggregation rules such as KRUM (Blanchard et al., 2017a), RFA (Pillutla et al., 2022), and the coordinate-wise median, when combined with their proposed *bucketing* technique, indeed satisfy Assumption 1. Thus, concrete and practical instantiations of robust aggregators are available within this framework. In addition, momentum-based or explicit MARINA/SVRG/SAGA periodic full-gradient (or large-minibatch) techniques (Gorbunov et al., 2023; Rammal et al., 2024) are necessary to achieve robustness against sophisticated attacks. While heavy-ball momentum itself can be interpreted as a form of variance reduction (Cutkosky & Orabona, 2019), throughout this paper we refer to heavy-ball-style updates simply as momentum. Without such techniques, Karimireddy et al. (2021) showed a fundamental lower bound demonstrating that learning fails when stochastic gradient noise is not properly controlled, making these methods essential for countering time-coupled attacks (Baruch et al., 2019).

**Federated Learning with Partial Participation** Federated learning with partial participation is a fundamental characteristic of practical federated learning systems. Real-world deployments inherently involve clients with heterogeneous capabilities and intermittent availability due to device constraints, battery limitations, and network connectivity variations (McMahan et al., 2017; Kairouz et al., 2021). This participation pattern directly impacts communication efficiency and system scalability, making it a critical consideration for algorithm design.

In the usual partial participation setting, all clients are assumed to be non-Byzantine, i.e., $\mathcal{G} = [n]$. The classical FEDAVG algorithm (McMahan et al., 2017) samples a subset of active clients, denoted by $\mathcal{S}_t \subseteq [n]$, uniformly at random at each round $t$, and aggregates their local updates by naive averaging: $\frac{1}{|\mathcal{S}_t|} \sum_{i \in \mathcal{S}_t} g_i^t$, where $g_i^t$ denotes the local gradient estimator of client $i$ (e.g., a stochastic gradient).

**Failure of Byzantine-robust Learning with Partial Participation** A natural extension of the full participation setting is to replace the naive averaging step

$$\frac{1}{|\mathcal{S}_t|} \sum_{i \in \mathcal{S}_t} g_i^t \quad \longrightarrow \quad \mathrm{Agg}(\{g_i^t\}_{i \in \mathcal{S}_t}).$$

While appealing, **this strategy fails with partial participation**: in some rounds, the sampled set may contain a Byzantine majority, despite the global condition $\delta < 1/2$. Under the (standard) model where the server only receives the gradients/momentum submitted in that round and has no additional side information, no (per-round) robust aggregator can reliably distinguish adversarial updates from honest updates when the sampled set contains a Byzantine majority. Furthermore, if we consider i.i.d. Bernoulli sampling, the likelihood of the existence of at least one round containing a Byzantine-majority grows exponentially with time.

Recent work has sought to address this issue. Allouah et al. (2024) provided lower bounds on the subsample size. However, due to a lack of momentum or large-minibatch sampling, their method collapses under time-coupled attacks such as ALIE (Baruch et al., 2019). Malinovsky et al. (2024) established convergence guarantees tolerating Byzantine-majority rounds via gradient-difference clipping, but their analysis relies on MARINA/SVRG/SAGA-style periodic full-gradient optimizers, which are known to be ineffective in deep learning (Defazio & Bottou, 2019).

## 4 PROPOSED METHOD

In this section, we propose **delayed momentum aggregation**, which is to apply the robust aggregator not only to the momentum of sampled clients but also to the cached momentum of non-sampled clients. Then, we propose a delayed momentum aggregation-based optimizer D-Byz-SGDM, which is Byzantine-robust even if only a subset of clients participate in each round. Formally, let $x^t$ denote the global model parameter maintained by the server at round $t$. The server then updates it using delayed momentum aggregation as follows:

$$x^t = x^{t-1} - \eta \, \mathrm{Agg}\left(\{m_i^t\}_{i \in \mathcal{S}_t} \cup \{m_i^{t-\tau(i,t)}\}_{i \in [n] \setminus \mathcal{S}_t}\right), \quad \text{(delayed momentum aggregation)}$$

where each $m_i^t$ represents a local momentum estimate, and $\tau(i,t)$ denotes the (possibly stochastic) delay since client $i$'s last update was received. This design maintains that $\mathrm{Agg}(\cdot)$ consistently sees the global Byzantine fraction $\delta < 1/2$, ensuring robustness even with partial participation.

As a concrete special case of the main idea, we propose a new method, D-Byz-SGDM, whose update rule is given in Algorithm 1. In each round $t$, the server independently samples each client with probability $p$ (i.e., $z^t \sim \mathrm{Ber}(p)^{\otimes n}$ and $\mathcal{S}_t = \{i : z_i^t = 1\}$). The selected clients refresh their momentum, while non-selected clients retain their cached value:

$$m_i^t = \begin{cases} (1-\alpha)m_i^{t-1} + \alpha \, \nabla f_i(x^{t-1}, \xi_i^{t-1}), & i \in \mathcal{S}_t, \\ m_i^{t-1}, & i \notin \mathcal{S}_t, \end{cases}$$

where $\alpha \in (0,1]$ is the client momentum parameter. Note that each client $i$ is included in $\mathcal{S}_t$ with probability $p$. Importantly, D-Byz-SGDM introduces no extra communication overhead. The server simply maintains one vector $m_i^t$ per client while reusing cached momentum for non-sampled clients, resulting in a memory requirement matching the full participation setting. As a possible mitigation for extreme cross-device regimes, one could explore streaming robust mean estimators (e.g., the streaming-based robust aggregator of Diakonikolas et al. (2022)) to shrink the server-side memory footprint, though such techniques are not yet compatible with our current robust aggregation definition, and we leave this integration for future work.

---

**Algorithm 1:** Optimizer with delayed momentum aggregation: D-Byz-SGDM

---

**Require:** initial vectors $x^0, m^0$, stepsize $\eta$, momentum parameter $\alpha$, robust aggregator Agg,
    client sampling probability $p \in (0, 1]$
Initialize $m_i^0$ and $\tau(i, 0) \leftarrow 0$ for all $i \in [n]$;
**for** $t = 1, 2, \ldots$ **do**
$\quad$ Sample $\mathcal{S}_t \subseteq [n]$ by including each $i \in [n]$ independently with prob. $p$
$\quad$ Server broadcasts $x^{t-1}$ to all $i \in \mathcal{S}_t$
$\quad$ **foreach** $i \in \mathcal{S}_t$ **in parallel do**
$\quad\quad$ Draw $\xi_i^{t-1} \sim \mathcal{D}_i$ and compute
$$m_i^t \leftarrow (1 - \alpha) m_i^{t-1} + \alpha \nabla F_i(x^{t-1}; \xi_i^{t-1})$$
$\quad\quad$ Send $m_i^t$ to server
$\quad$ **end**
$\quad$ **foreach** $i \notin \mathcal{S}_t$ *(on server)* **do**
$\quad\quad$ Update $m_i^t \leftarrow m_i^{t-1}$
$\quad$ **end**
$\quad$ $m^t \leftarrow \text{Agg}\Big(\{m_i^t\}_{i \in \mathcal{S}_t} \cup \{m_i^t\}_{i \notin \mathcal{S}_t}\Big)$ `// delayed momentum aggregation`
$\quad$ $x^t \leftarrow x^{t-1} - \eta\, m^t$
**end**

---

## 5 EXPERIMENTS

We evaluated D-Byz-SGDM under various Byzantine attacks with partial participation ($p = 0.5$) by training a convolutional network on MNIST and a ResNet-18 on CIFAR-10 across IID and non-IID data partitions. We compared four optimizers (FedAvg, FedAvgM, D-Byz-SGDM, and the heuristic momentum extension of Byz-VR-MARINA-PP from Malinovsky et al. (2024)) with five robust aggregators under six Byzantine attacks. FedAvg (McMahan et al., 2017) performed single-step SGD per client followed by server-side aggregation, while FedAvgM (Cheng et al., 2024) extended this with client-side momentum ($\beta = 0.9$). In our setting, the standard averaging step in four optimizers was replaced by robust aggregation rules, allowing us to assess performance under Byzantine attacks. Our implementation extended Karimireddy et al. (2022)'s codebase[1] with attacks from the ByzFL framework (González et al., 2025) and additional support for CIFAR-10/ResNet-18 training. Appendix C provided complete experimental details.

**Hyperparameter selection.** For each optimizer (FedAvg, FedAvgM, D-Byz-SGDM) we tuned a global learning rate $\eta$ over the grid $\{0.1, 0.01, 0.001\}$. Byz-VR-MARINA-PP required tuning both $\eta$ and the clipping radius $\lambda \in \{10.0, 1.0, 0.1\}$. Every configuration was evaluated over seeds $\{0, 1, 2\}$, and we selected the setting with the highest mean validation accuracy for reporting in both the non-Byzantine and Byzantine settings.

### 5.1 BYZANTINE ROBUSTNESS WITH PARTIAL PARTICIPATION (MAIN RESULT)

We analyzed partial participation ($p = 0.5$) with $n = 25$ total clients of which 20% were Byzantine ($\delta = 0.2$). All plots in this subsection used centered clipping (CCLIP) (Karimireddy et al., 2021) as the server-side aggregator.

**Key findings.** Figures 1a and 1b demonstrate the performance of algorithms with the CCLIP aggregator under Byzantine attacks with partial participation ($p = 0.5$). Our experiments reveal three critical insights: (1) D-Byz-SGDM *consistently achieved the highest final accuracy across all settings.* On MNIST IID (upper half of Fig. 1a), both D-Byz-SGDM and Byz-VR-MARINA-PP achieved near-perfect accuracy, while FedAvg and FedAvgM diverged after three epochs. On CIFAR-10 with ResNet-18 (upper half of Fig. 1b), D-Byz-SGDM sustained 80–85% accuracy across all attack types. (2) *Non-IID data exposed critical algorithmic differences.* On non-IID

---

[1] https://github.com/epfml/byzantine-robust-noniid-optimizer

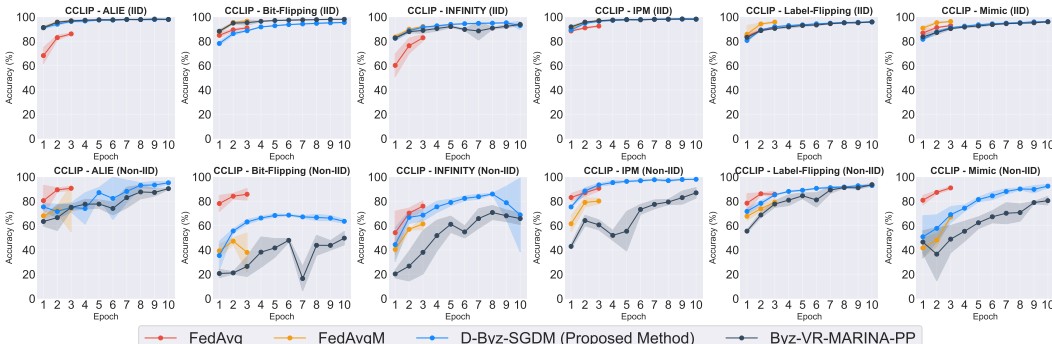

(a) MNIST under centered clipping (CCLIP). The top row shows IID splits and the bottom row shows non-IID splits with bucketing $s = 2$; from left to right the columns correspond to ALIE, Bit-Flipping, INFINITY, IPM, Label-Flipping, and Mimic attacks. *Observation:* D-Byz-SGDM remains the most stable and accurate across all attacks, while FedAvg/FedAvgM diverge when many Byzantines are sampled.

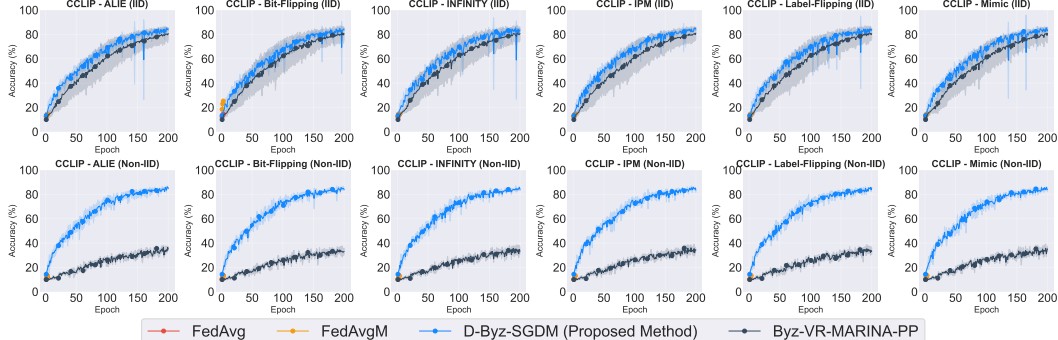

(b) CIFAR-10 (ResNet-18) under centered clipping (CCLIP). The top row reports IID splits and the bottom row reports non-IID splits with bucketing $s = 2$; moving left to right the columns correspond to ALIE, Bit-Flipping, INFINITY, IPM, Label-Flipping, and Mimic attacks. *Observation:* D-Byz-SGDM sustains 80–85% accuracy across attacks, whereas FedAvg/FedAvgM often collapse by epoch $\approx 4$ when a Byzantine majority is sampled.

Figure 1: Byzantine-robust training with CCLIP and partial participation ($p = 0.5$). D-Byz-SGDM is consistently the most accurate and stable curve.

MNIST (lower half of Fig. 1a), Byz-VR-MARINA-PP exhibited high variance and unstable convergence, while D-Byz-SGDM maintained consistent performance. The disparity was dramatic on non-IID CIFAR-10 (lower half of Fig. 1b): Byz-VR-MARINA-PP catastrophically failed (20–35% accuracy), whereas D-Byz-SGDM maintained 80–85% accuracy. The delayed momentum aggregation principle proved crucial. While standard methods failed when a Byzantine majority was sampled,[2] D-Byz-SGDM maintained stable convergence. (3) *The approach generalizes across aggregators.* Similar trends held across other aggregators (avg, krum, cm, rfa) and both datasets, with FedAvg and FedAvgM performing poorly in both IID and non-IID settings (FedAvgM showed marginal improvements only in specific attacks like Bit-Flipping); see Appendix D for the full set of figures.

## 5.2 BASELINE PERFORMANCE WITHOUT BYZANTINE CLIENTS

We also examined the non-Byzantine setting ($\delta = 0$) to establish baseline performance. The setup used $n = 20$ clients with the avg aggregator. The results were summarized in Figures 2a and 2b.

**Key findings.** Across both IID and non-IID settings on MNIST *(Fig. 2a), Byz-VR-MARINA-PP achieved the worst validation accuracy and highest loss throughout training*. Surprisingly, D-Byz-SGDM *consistently outperformed* FedAvgM *in the non-Byzantine setting* ($\delta = 0$), despite the risk that reusing momentum across rounds could degrade performance. The advantage persisted on the

---

[2]With $p = 0.5$, if many Byzantines were sampled together, they could overwhelm the aggregation.

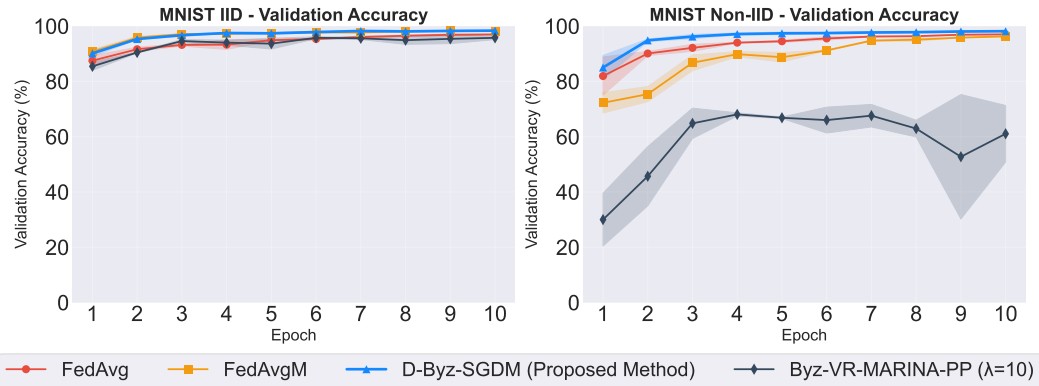

(a) MNIST without Byzantine clients ($\delta = 0$) using the `avg` aggregator. The left panel reports the IID partition and the right panel reports the non-IID partition. *Observation:* performance is saturated in IID; D-Byz-SGDM retains a clear margin in non-IID, showing delayed momentum aggregation mitigates heterogeneity even without attacks.

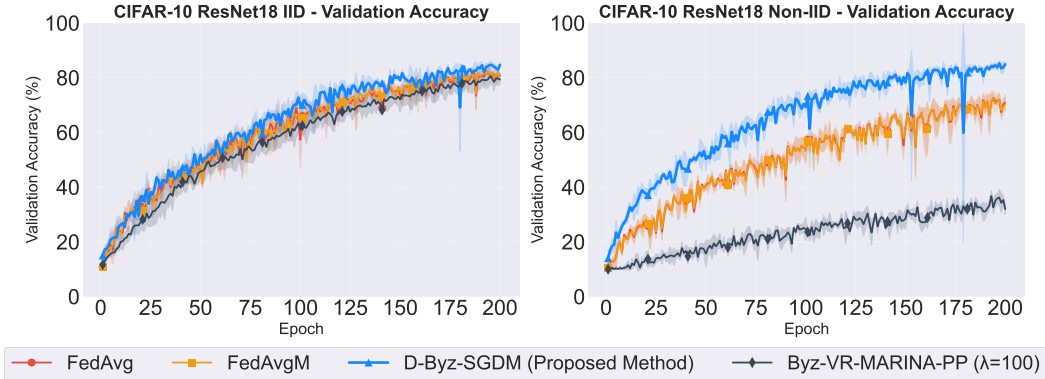

(b) CIFAR-10 / ResNet-18 without Byzantine clients ($\delta = 0$) using the `avg` aggregator. The left panel shows the IID partition and the right panel shows the non-IID partition. *Observation:* D-Byz-SGDM converges faster and finishes 5–10 points higher on both partitions, while Byz-VR-MARINA-PP remains well below momentum baselines.

Figure 2: Baseline training with partial participation ($p = 0.5$) and no Byzantine clients ($\delta = 0$). Delayed momentum aggregation (D-Byz-SGDM) remains strongest in non-IID even without adversaries.

deeper model (ResNet-18) on CIFAR-10 (Fig. 2b), underscoring that delayed momentum aggregation scaled beyond vision tasks with shallow networks. The curves suggested that with partial participation ($p = 0.5$) and heterogeneity (non-IID), the delayed momentum aggregation mechanism in D-Byz-SGDM mitigated heterogeneity-induced drift, acting as an *implicit regularizer* even without attacks. We further examined Byz-VR-MARINA-PP in the non-Byzantine regime. Somewhat unexpectedly, applying clipping to momentum differences introduced a *bias* detrimental to performance unless the clipping hyperparameter $\lambda$ was chosen with extreme care. This sensitivity highlighted a trade-off: while clipping was essential to defend against Byzantine behaviors, it could significantly distort gradient estimates in non-Byzantine settings.

## 6 CONCLUSION

We proposed *delayed momentum aggregation*, a novel principle where servers aggregate fresh momentum from participating clients with the most recently received momentum from non-participating clients. Our D-Byz-SGDM optimizer maintains Byzantine-robustness under partial participation while remaining lightweight to deploy. Experiments showed consistent improvements over existing methods across various attacks and data distributions. The delayed momentum aggregation principle opens promising avenues for extension to other client selection schemes (Fraboni

et al., 2022; Cho et al., 2020; Fraboni et al., 2021; Li et al., 2020; Chen et al., 2022) beyond Bernoulli sampling.

## ETHICS STATEMENT

This work addresses robustness in federated learning against adversarial participants. We use "Byzantine" following established distributed systems nomenclature to denote arbitrary failures, with no cultural reference intended. Our proposed method is defensive, designed to enhance the reliability and safety of collaborative training.

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

## A   LARGE LANGUAGE MODEL (LLM) USAGE

In accordance with ICLR 2026 guidelines, we disclose the use of Large Language Models (LLMs) in preparing this submission:

- **Writing assistance:** Yes, LLMs were used to aid and polish writing. Specifically, we employed LLMs to improve clarity, correct grammar, and enhance the overall presentation of technical content throughout the manuscript. The authors take full responsibility for all content, including any LLM-assisted portions.
- **Literature retrieval and discovery:** Yes, LLMs were used for finding related work. We utilized LLMs to help identify relevant papers, understand connections between different research areas, and ensure comprehensive coverage of the Byzantine-robust federated learning literature. All citations were independently verified for accuracy.

We emphasize that all research ideas, algorithm design, experimental design, and results analysis were conducted by the authors without LLM involvement. The LLMs served solely as auxiliary tools for improving presentation and literature discovery.

## B   ALGORITHM DETAILS

We present the detailed algorithm for D-Byz-SGDM (Delayed Byzantine-robust SGD with Momentum), which implements our delayed momentum aggregation principle. The key idea is to apply the robust aggregator not only to the momentum of sampled clients but also to the cached momentum of non-sampled clients, ensuring that the aggregator consistently sees the global Byzantine fraction $\delta < 1/2$ even under partial participation.

In each round $t$, the server independently samples each client with probability $p$ (i.e., $z^t \sim \mathrm{Ber}(p)^{\otimes n}$ and $\mathcal{S}_t = \{i : z_i^t = 1\}$). The selected clients refresh their momentum using:

$$m_i^t = \begin{cases} (1-\alpha)m_i^{t-1} + \alpha \, \nabla f_i(x^{t-1}, \xi_i^{t-1}), & i \in \mathcal{S}_t, \\ m_i^{t-1}, & i \notin \mathcal{S}_t, \end{cases}$$

where $\alpha \in (0, 1]$ is the client momentum parameter. Non-selected clients retain their cached momentum values from previous rounds.

The server then performs delayed momentum aggregation by applying the robust aggregator Agg to the union of fresh momentum from sampled clients and cached momentum from non-sampled clients:

$$m^t = \mathrm{Agg}\Big(\{m_i^t\}_{i \in \mathcal{S}_t} \cup \{m_i^t\}_{i \notin \mathcal{S}_t}\Big)$$

This design ensures that even when partial participation might lead to a Byzantine majority among sampled clients, the aggregator always operates on the full set of clients (fresh and cached), maintaining robustness.

To see how this corresponds to the delayed momentum aggregation principle, note that the delay function $\tau(i, t)$ represents the number of rounds since client $i$'s momentum was last updated. Formally:

$$\tau(i, t) = \min\{s \geq 0 : i \in \mathcal{S}_{t-s}\}$$

This is a random variable that depends on the sampling history. When $i \in \mathcal{S}_t$, we have $\tau(i, t) = 0$ (fresh update), and when $i \notin \mathcal{S}_t$, we have $\tau(i, t) > 0$ (stale update). The algorithm effectively implements:

$$x^t = x^{t-1} - \eta \, \mathrm{Agg}\Big(\{m_i^t\}_{i \in \mathcal{S}_t} \cup \{m_i^{t-\tau(i,t)}\}_{i \in [n] \setminus \mathcal{S}_t}\Big)$$

where for non-sampled clients, $m_i^{t-\tau(i,t)}$ is their most recent momentum update, which is exactly what we store as $m_i^t$ in the algorithm.

Importantly, D-Byz-SGDM does not incur additional communication costs compared to standard partial participation methods: the server only queries sampled clients and stores one momentum vector $m_i^t$ per client, matching the memory requirements of full participation settings.

## C  ADDITIONAL EXPERIMENTAL DETAILS

### C.1  COMMON EXPERIMENTAL SETTINGS

All experiments covered two vision workloads: MNIST with a convolutional neural network architecture (CONV-CONV-DROPOUT-FC-DROPOUT-FC) and CIFAR-10 with a standard ResNet-18. Training employed cross-entropy (negative log-likelihood) loss with batch size 32 per client and client participation probability $p = 0.5$. We evaluated both IID and non-IID data partitions, with the latter following the class-based approach of Karimireddy et al. (2022). Four optimizers were compared: FedAvg, FedAvgM, D-Byz-SGDM, and the heuristic momentum extension of Byz-VR-MARINA-PP (with $\lambda \in \{10.0, 1.0, 0.1\}$) introduced in (Malinovsky et al., 2024), all using momentum parameter $\alpha = 0.9$ where applicable. Training ran for 10 epochs (300 iterations total) for MNIST and 200 epochs for CIFAR-10, with results averaged over seeds $\{0, 1, 2\}$. For each optimizer we tuned the learning rate $\eta \in \{0.1, 0.01, 0.001\}$; additionally Byz-VR-MARINA-PP tuned the clipping radius $\lambda \in \{10.0, 1.0, 0.1\}$. We selected the configuration with the highest mean validation accuracy across the three seeds for both the non-Byzantine and Byzantine experiments. Tables 1–4 provided complete configuration details.

### C.2  BASELINE PERFORMANCE EVALUATION

This experiment established baseline performance under partial participation without Byzantine clients across both MNIST (ConvNet) and CIFAR-10 (ResNet-18). We used $n = 20$ clients with no Byzantine clients ($\delta = 0$) and naive averaging aggregation. The objective was to validate that D-Byz-SGDM maintains competitive performance in non-Byzantine settings and to establish reference performance levels for subsequent robustness comparisons. Results in Figs. 2a and 2b demonstrated that D-Byz-SGDM outperformed standard momentum methods on both MNIST and CIFAR-10 even without adversaries, suggesting that delayed momentum aggregation provided implicit regularization benefits under heterogeneous data distributions.

### C.3  BYZANTINE ROBUSTNESS ASSESSMENT

This experiment evaluated robustness against Byzantine attacks under partial participation on both datasets (MNIST with the ConvNet backbone and CIFAR-10 with ResNet-18). We configured $n = 25$ clients with 5 Byzantine clients (20%). Five robust aggregators were evaluated: Krum, coordinate-wise median, CCLIP (centered clipping), RFA, and naive averaging as baseline. The experimental design included both IID and non-IID data partitions, with bucketing applied in the Byzantine non-IID setting to mitigate extreme heterogeneity. This comprehensive evaluation spanned 6,480 total experimental runs across all combinations of attacks, aggregators, optimizers, data partitions, and random seeds (3,240 runs per dataset).

### C.4  NON-IID DATA PARTITION

We constructed the non-IID split following Karimireddy et al. (2022) in the *balanced* case: (i) sorted the training sets by label; (ii) split it into $G$ equal, contiguous shards (where $G$ is the number of good/honest clients); (iii) assigned one shard to each honest client and shuffle examples within each client. We partitioned the test set analogously.

### C.5  COMPUTING ENVIRONMENT

Experiments ran on NVIDIA A100-SXM4-80GB GPUs (CUDA 12.2) and AMD EPYC 7763 CPUs. Table 5 provides detailed hardware and software specifications.

## D  EXTENDED RESULTS

**Per-aggregator curves with Byzantine clients.**  This section complemented Figs. 1a and 1b by showing training dynamics for the other robust aggregators across the same attacks, data partitions, and optimizers on MNIST (ConvNet) and CIFAR-10 (ResNet-18).

Table 1: MNIST (non-Byzantine) configuration used in Fig. 2a.

| | |
|---|---|
| Dataset | MNIST (IID and non-IID partitions) |
| Model | CONV-CONV-DROPOUT-FC-DROPOUT-FC |
| Clients | $n = 20$ (all honest) |
| Participation | $p = 0.5$ (partial participation) |
| Aggregator | `avg` |
| Batch size | 32 per client |
| Training horizon | 10 epochs (300 rounds) |
| Optimizers | FedAvg, FedAvgM, D-Byz-SGDM, Byz-VR-MARINA-PP |
| Learning-rate tuning | grid search on $\{0.1, 0.01, 0.001\}$ |
| Byz-VR-MARINA-PP tuning | joint grid search $\eta \in \{0.1, 0.01, 0.001\}, \lambda \in \{10.0, 1.0, 0.1\}$ |
| Seeds | $\{0, 1, 2\}$ |
| Attacks | none |

Table 2: MNIST (Byzantine) configuration used in Fig. 1a.

| | |
|---|---|
| Dataset | MNIST (IID and non-IID with bucketing $s = 2$) |
| Model | CONV-CONV-DROPOUT-FC-DROPOUT-FC |
| Clients | $n = 25$ (20 honest, 5 Byzantine; $\delta = 0.2$) |
| Participation | $p = 0.5$ (partial participation) |
| Aggregators | `avg`, `krum`, `cm`, `CCLIP`, `rfa` |
| Batch size | 32 per client |
| Training horizon | 10 epochs (300 rounds) |
| Attacks | BF, LF, mimic, IPM, ALIE, INF |
| Optimizers | FedAvg, FedAvgM, D-Byz-SGDM, Byz-VR-MARINA-PP |
| Learning-rate tuning | grid search on $\{0.1, 0.01, 0.001\}$ |
| Byz-VR-MARINA-PP tuning | joint grid search $\eta \in \{0.1, 0.01, 0.001\}, \lambda \in \{10.0, 1.0, 0.1\}$ |
| Seeds | $\{0, 1, 2\}$ |

*Notation:* `avg`=naive average, `krum`=Krum (Blanchard et al., 2017a), `cm`=coordinate-wise median, `CCLIP`=centered clipping (Karimireddy et al., 2021), `rfa`=geometric median (RFA) (Pillutla et al., 2022).

Table 3: CIFAR-10 (non-Byzantine) configuration used in Fig. 2b.

| | |
|---|---|
| Dataset | CIFAR-10 (IID and non-IID partitions) |
| Model | ResNet-18 |
| Clients | $n = 20$ (all honest) |
| Participation | $p = 0.5$ (partial participation) |
| Aggregator | `avg` |
| Batch size | 32 per client |
| Training horizon | 200 epochs |
| Optimizers | FedAvg, FedAvgM, D-Byz-SGDM, Byz-VR-MARINA-PP |
| Learning-rate tuning | grid search on $\{0.1, 0.01, 0.001\}$ |
| Byz-VR-MARINA-PP tuning | joint grid search $\eta \in \{0.1, 0.01, 0.001\}$, $\lambda \in \{10.0, 1.0, 0.1\}$ |
| Seeds | $\{0, 1, 2\}$ |
| Attacks | none |

Table 4: CIFAR-10 (Byzantine) configuration used in Fig. 1b.

| | |
|---|---|
| Dataset | CIFAR-10 (IID and non-IID with bucketing $s = 2$) |
| Model | ResNet-18 |
| Clients | $n = 25$ (20 honest, 5 Byzantine; $\delta = 0.2$) |
| Participation | $p = 0.5$ (partial participation) |
| Aggregators | `avg`, `krum`, `cm`, `CCLIP`, `rfa` |
| Batch size | 32 per client |
| Training horizon | 200 epochs |
| Attacks | BF, LF, mimic, IPM, ALIE, INF |
| Optimizers | FedAvg, FedAvgM, D-Byz-SGDM, Byz-VR-MARINA-PP |
| Learning-rate tuning | grid search on $\{0.1, 0.01, 0.001\}$ |
| Byz-VR-MARINA-PP tuning | joint grid search $\eta \in \{0.1, 0.01, 0.001\}$, $\lambda \in \{10.0, 1.0, 0.1\}$ |
| Seeds | $\{0, 1, 2\}$ |

*Notation:* `avg`=naive average, `krum`=Krum (Blanchard et al., 2017a), `cm`=coordinate-wise median, `CCLIP`=centered clipping (Karimireddy et al., 2021), `rfa`=geometric median (RFA) (Pillutla et al., 2022).

Table 5: Runtime hardware and software.

| | |
|---|---|
| **CPU** | |
| Model name | AMD EPYC 7763 64-Core Processor |
| # CPU(s) | 128 |
| **GPU** | |
| Product Name | NVIDIA A100-SXM4-80GB |
| CUDA Version | 12.2 |
| **PyTorch** | |
| Version | 2.7.1 |

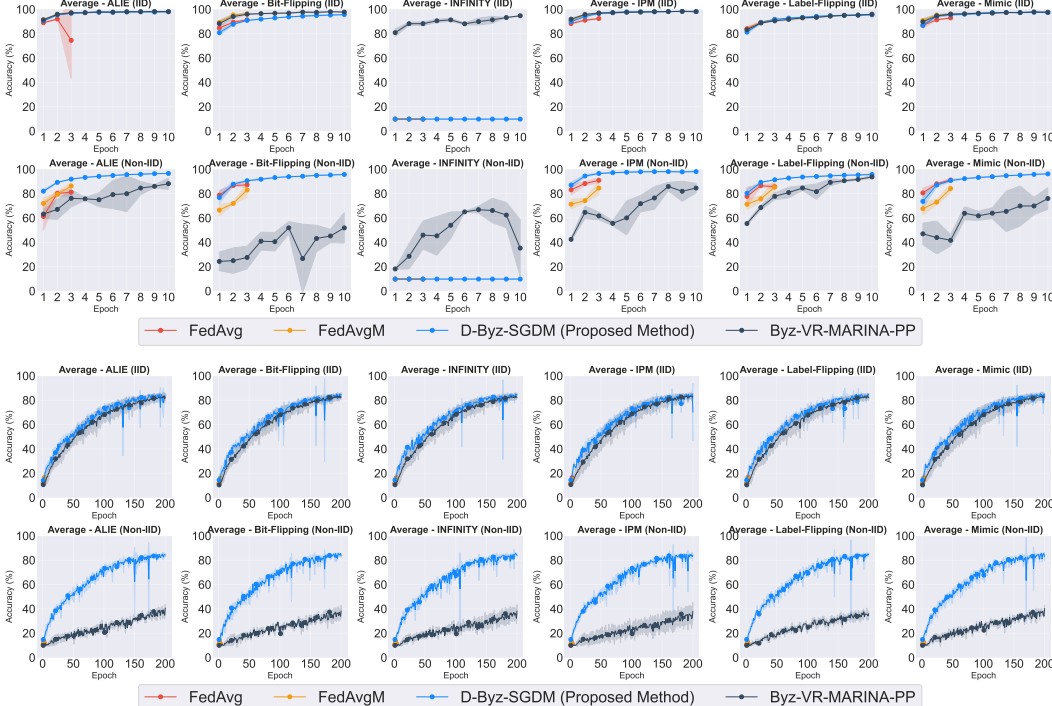

Figure 3: `avg` (naive average) under Byzantine attacks ($p = 0.5$). The top row presents MNIST and the bottom row presents CIFAR-10; within each row the first strip is IID and the second strip is non-IID with bucketing $s = 2$. Columns proceed left to right through ALIE, Bit-Flipping, INFINITY, IPM, Label-Flipping, and Mimic.

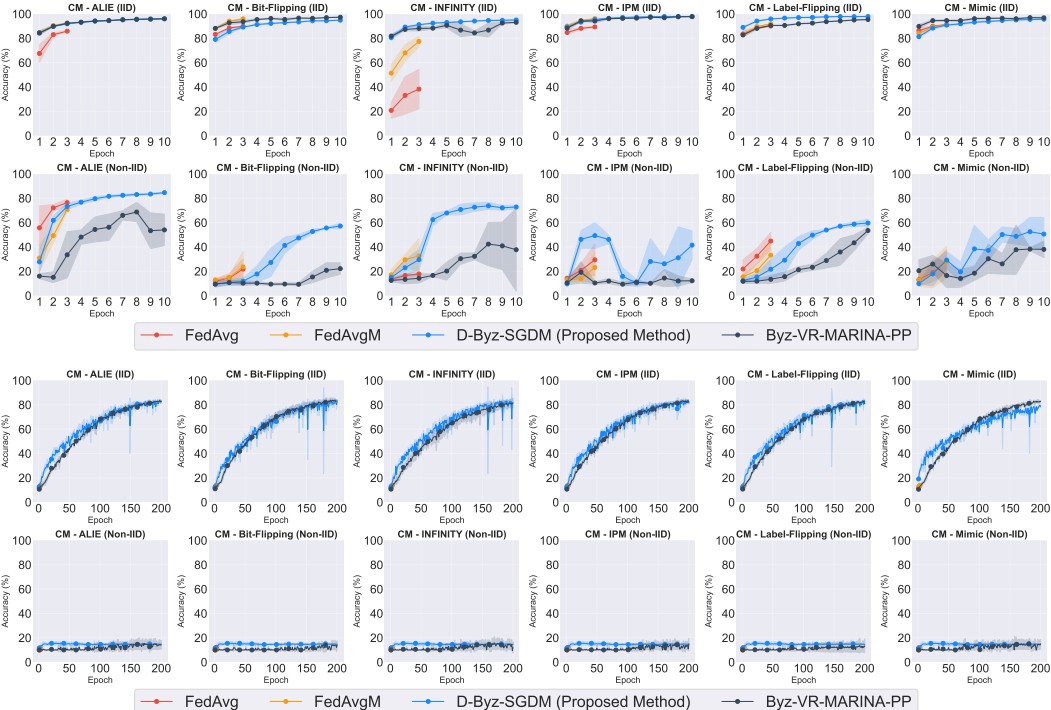

Figure 4: cm (coordinate-wise median) under Byzantine attacks ($p = 0.5$). The layout matches Fig. 3: top row MNIST, bottom row CIFAR-10; within each row an IID strip is followed by a non-IID strip ($s = 2$); columns move left to right through ALIE, Bit-Flipping, INFINITY, IPM, Label-Flipping, and Mimic.

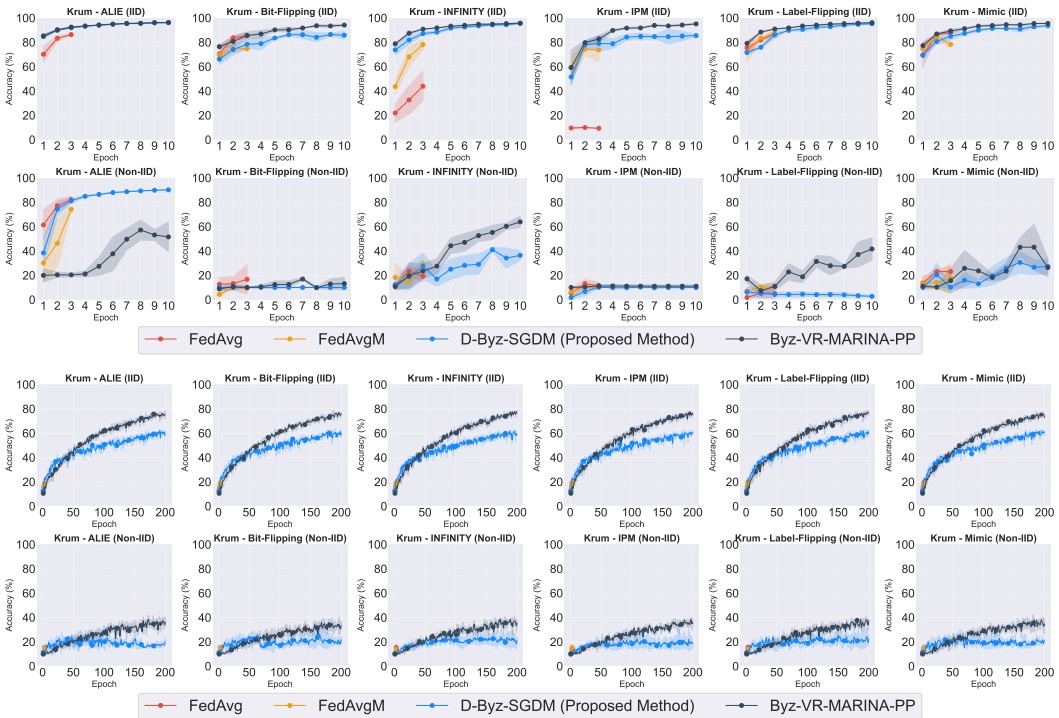

Figure 5: krum / Multi-Krum under Byzantine attacks ($p = 0.5$). The top row shows MNIST and the bottom row shows CIFAR-10; each row contains an IID strip followed by a non-IID strip with bucketing $s = 2$. Columns list ALIE, Bit-Flipping, INFINITY, IPM, Label-Flipping, and Mimic.

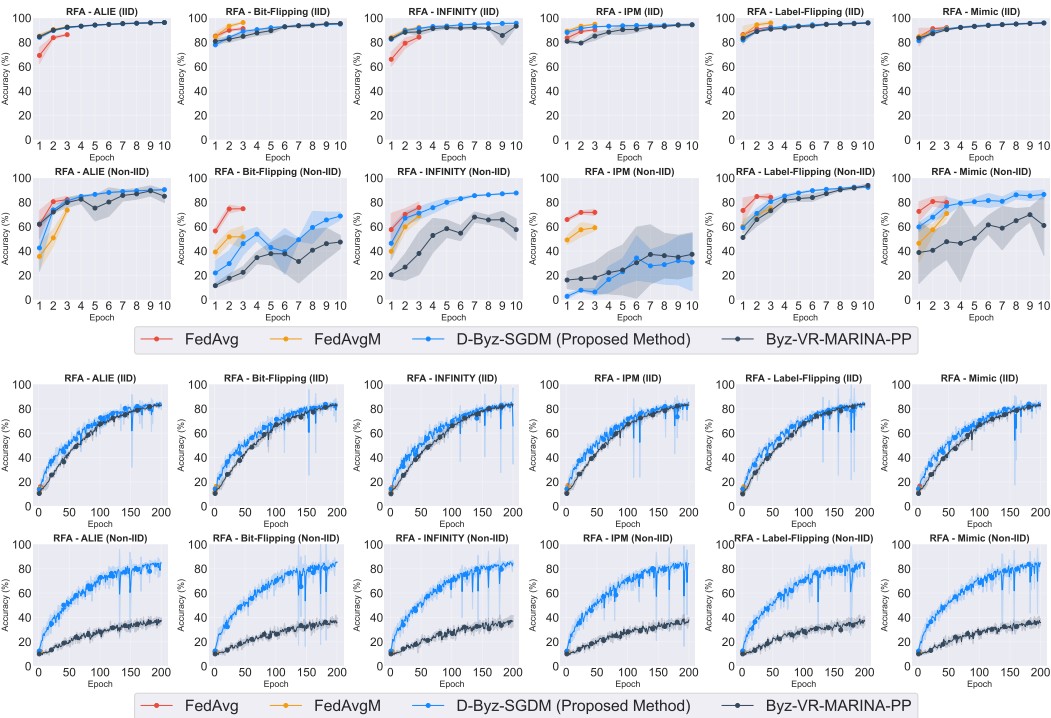

Figure 6: rfa (Robust Federated Averaging) under Byzantine attacks ($p = 0.5$). The top row covers MNIST and the bottom row covers CIFAR-10; within each row an IID strip precedes a non-IID strip with bucketing $s = 2$. Columns run left to right through ALIE, Bit-Flipping, INFINITY, IPM, Label-Flipping, and Mimic.

