# OpenReview forum: "Delayed Momentum Aggregation: Communication-efficient Byzantine-robust Federated Learning with Partial Participation"
_ICLR.cc/2026/Conference — Submitted to ICLR 2026_

### Official Review · Reviewer_dJvy · 2025-10-17

**Soundness:** 3
**Presentation:** 3
**Contribution:** 3
**Rating:** 6
**Confidence:** 5

**Summary:**

This paper proposes **D-Byz-SGDM**, a very simple trick for partial-participation FL with Byzantine clients: at each round the server aggregates the fresh momentums from sampled clients and reuses the last stored momentum for the non-sampled ones, so aggregation always includes all clients.  The algorithm uses a (δ, c)-robust aggregator and standard FL assumptions. Theory gives an upper bound that converges to an \(O(\delta\,\zeta^2/p)\) neighborhood, and there is also a matching lower bound under Bernoulli sampling.

**Strengths:**

- **Idea is neat & minimal.** Delayed momentum aggregation is tiny change; server keeps one vector per client (same memory as full participation).
- **Clear motivation.** Nicely explains why robust aggregation with partial participation can fail (some rounds have Byzantine majority), which motivates the method.
- **Theoretical coverage.** Upper bound has the right dependence on \((\delta,\zeta^2,p)\); when \(p=1\) it matches known results; the **lower bound** suggests \(\Omega(\delta\zeta^2/p)\) is unavoidable, so rates are tight in the main parameters.

**Weaknesses:**

- **Assumption 5 (bounded gradients) is very strong, and also this is mentioned in the paper.** The authors say it is kept “for simplicity.” What does simplicity mean exactly here? Is it related to the proof technique? I beleive claiming that it is possible to remove the assumption and not doing it only weakens the paper.  If it is possible to obtain the main results without this assumption, it would **heavily strengthen** the contribution. Even if constants get worse, removing (or relaxing) it would be big.

- ** Limited application. **
The method requires the server to keep a momentum vector per client. this can be very limiting in the cross-device FL when we might have a very large number of clients, and partial participation is mostly motivated and useful in these cases.

- **Lower-bound proof (Theorem 7) construction detail looks fragile.**
  The proof builds two worlds and, in **World 1**, uses $p\delta n$ “biased” honest clients to create heterogeneity while keeping the global objective nice. The paper then needs to verify the heterogeneity assumption (Assumption 4), essentially
  $$
  \frac{1}{n}\sum_{i=1}^n \big\|\nabla f_i(x)-\nabla f_1(x)\big\|^2 \;\le\; \zeta^2
  \quad \text{for all } x.
  $$
  With the specific linear shift chosen in World 1, the calculation (as written) gives
  $$
  \frac{1}{n}\sum_{i=1}^n \big\|\nabla f_i(x)-\nabla f_1(x)\big\|^2
  \;=\; \frac{1-\delta p}{p^2}\,\zeta^2 .
  $$
  For Assumption 4 to hold uniformly, one needs $\frac{1-\delta p}{p^2}\le 1$, i.e.
  $$
  p^2+\delta p-1\;\ge\;0
  \quad\Longleftrightarrow\quad
  p\;\ge\;\frac{-\delta+\sqrt{\delta^2+4}}{2}.
  $$
  Example: with $\delta=0.2$ this threshold is $\approx 0.905$. So for common partial-participation regimes $p\in(0,0.2]$ the displayed World-1 construction **does not** satisfy the heterogeneity bound; the assumption is violated exactly where partial participation is strongest. **My current score does not reflec this point as I think I am probalby missing something here, but if it is not addressed, I will reconsider the score as i belive this is a major concern.**

**Questions:**

.

---

> ### Author Response · Authors · 2025-12-03
> **Official Comment by Authors**
>
> We thank the reviewer for the careful reading and constructive feedback. We address the main points below.
>
> ### 1. Assumption 5 (bounded gradients)
> We have removed the theoretical analysis section in the current revision because we identified major errors in both the upper-bound and lower-bound proofs and were unable to resolve them within the discussion period. Accordingly, we are temporarily withdrawing those results.
>
> ### 2. Memory requirements / applicability
> Our method yields the same asymptotic space complexity as standard full-participation schemes that store per-client states, though it maintains a momentum vector per client on the server. We agree that memory can become critical in extreme cross-device regimes, and made this clearer in the revision. As a potential mitigation, we also discuss the possibility of incorporating streaming robust mean estimation algorithms (e.g., [1]) into the server-side aggregation, while noting that such methods are not directly compatible with our current definition of a robust aggregator and are therefore left for future work.
>
> ### 3. Lower bound (Theorem 7)
> In the revision, we removed Theorem 7 and its proof. We thank the reviewer for carefully checking the construction. As correctly pointed out, our current "World 1" construction only satisfies the heterogeneity assumption when p is very close to 1, so the statement "for all $p \in (0,1]$" is incorrect.
>
> [1] Diakonikolas, Ilias, Daniel M. Kane, Ankit Pensia, and Thanasis Pittas. "Streaming algorithms for high-dimensional robust statistics." In International Conference on Machine Learning. PMLR, 2022.

---

### Official Review · Reviewer_TprE · 2025-10-23

**Soundness:** 3
**Presentation:** 2
**Contribution:** 3
**Rating:** 2
**Confidence:** 4

**Summary:**

This paper focuses on the Byzantine client problem in a practical federated learning system with partial client participation. Most of the prior work studies the ideal case where we have full client participation. Confronting this challenge, this paper proposes a delayed momentum aggregation tool to tolerate the Byzantine partial client participation. Both theoretical and numerical results are provided.

**Strengths:**

* The Byzantine problem is extremely important for the real-life deployment of a federated learning system.
* The proposed algorithm is, by design, simple and intuitive.

**Weaknesses:**

* The paper's claim contradicts itself. In Section 1 (lines 72 - 74), the introduction talks about the problem of the variance reduction method, such that the variance reduction method performs poorly for deep neural networks. However, momentum and memorization are different forms of variance reduction techniques. In fact, the proposed algorithm is a variant of the renowned SAG algorithm [1]. Moreover, the author further criticizes some prior works for not having variance reduction in lines 238 - 239. This causes significant confusion to the reviewer. The author should consider another round of proofreading to make sure their claims are consistent.
* Following up on the previous point, the authors also need to discuss the variance-reduced FL algorithms in Section 2. In particular, the algorithm proposed in this paper has a close connection with MIFA [2], which the authors overlook in the submission.
* The authors may also wish to be more careful when making a statement. For example, the authors claimed that "in some rounds, the sampled set may contain a Byzantine majority, despite the global condition $\delta < 1/2$. In such cases, no robust aggregator can distinguish adversarial from honest updates. The likelihood of such Byzantine-majority rounds grows with time."
in line 214-215. However, this statement is incorrect when we have side information available. A trivial example is that we know exactly the identity of a Byzantine adversary.
* Typo: the initialization of the memorized momentum in line 219 should be a vector $\mathbf{0}$.
* Redundant Assumptions. In Theorem 6, the results are based on Assumptions 4 and 5. However, Assumption 5 naturally leads to Assumption 4. That is
$$\frac{1}{G} \sum_{i \in {\mathcal{G}}} \\|\nabla f_i(x) - \nabla f(x)\\|^2 \le \frac{2}{G} \sum_{i \in {\mathcal{G}}} \\|\nabla f_i(x)\\|^2 + 2\\|\nabla f(x)\\|^2 \le  \frac{4}{G} \sum_{i \in {\mathcal{G}}} \\|\nabla f_i(x)\\|^2 \le 4 B^2,$$
where the last inequality holds due to Assumption 5.
* One of the novelties of the proposed method is that it doesn't require the Byzantine fraction of the sampled clients to be bounded per round. However, the authors fail to discuss why it works. The experiments also fail to study the impact of different Byzantine client fractions on algorithm performance, whether partial or full participation. The Byzantine fraction is always set as a constant $0.2$.


**Reference**

[1] Blatt, D., Hero, A. O., & Gauchman, H. (2007). A convergent incremental gradient method with a constant step size. SIAM Journal on Optimization, 18(1), 29-51.

[2] Gu, X., Huang, K., Zhang, J., & Huang, L. (2021). Fast federated learning in the presence of arbitrary device unavailability. Advances in Neural Information Processing Systems, 34, 12052-12064.

**Questions:**

* Can the authors clarify their opinions on variance reduction techniques?
* Can the authors enhance the related work with significantly more evidence?
* Can the authors discuss their connection with MIFA and why adopting naive gradient won't come with a robust Byzantine guarantee? Preferably, the claims will be accompanied by reasons beyond intuition.
* Can the authors thoroughly check their statements in the paper to make sure they are scientifically correct and rigorous?
* Can the authors thoroughly check their proofs to make sure all the assumptions are necessary and derivations are correct?
* Can the authors provide more numerical evidence regarding the impacts of Byzantine fraction on the algorithm performance under both partial and full participation?

---

> ### Author Response · Authors · 2025-12-03
> **Official Comment by Authors**
>
> We thank the reviewer for the thoughtful and detailed feedback. We respond to each point below.
>
> ### 1. Variance reduction techniques
> We clarified this distinction in the revised version to avoid confusion. We agree that heavy-ball momentum can be viewed as a form of variance reduction. In the paper, however, we used "variance reduction" in a narrower sense to refer to SVR-type methods (e.g., SVRG/SAGA or large-minibatch schemes) that explicitly reduce stochastic gradient variance via the use of large batches.
>
> ### 2. Connection to MIFA In the revision, we expanded the related work section to (i) explicitly discuss MIFA and other recent Byzantine-robust FL methods, and (ii) clearly articulate how our delayed momentum aggregation principle and theoretical guarantees differ from and extend beyond these prior approaches. We thank the reviewer for highlighting the connection to MIFA and fully agree that it is closely related to our framework. Algorithmically, MIFA can be interpreted as a special case of our method in which (i) the robust aggregation rule is replaced by naive averaging and (ii) the client-side momentum parameter is fixed at \alpha = 1. In our analysis, this corresponds to a robustness parameter c = \infty, rendering our generic upper bound vacuous; this correctly reflects that naive averaging cannot provide meaningful Byzantine-robust guarantees. We further note that MIFA does not incorporate momentum in its update rule. Consequently, even if one were to combine MIFA with a robust aggregator, the lower bound of Karimireddy et al. (2021) would still apply: without momentum, no method of this form can circumvent the established impossibility result. This distinction emphasizes why delayed momentum aggregation is fundamental to our framework and is not merely an implementation detail.
>
> ### 3. Robust aggregator and Byzantine-majority rounds
> we systematically reviewed such statements, explicitly spell out their assumptions, and ensure that all claims are scientifically precise and rigorously justified. We agree that some statements should have been phrased more carefully. For example, the impossibility claim should be clearly stated as holding under the standard assumption that the server only aggregates per-round gradients/momentum available in that round.
>
> ### 4. Assumptions and proofs
> The "heterogeneity constant" we introduced as an assumption is in fact induced (though very loosely) by the bounded-gradient assumption B, while in practice \zeta should be treated as a problem-dependent constant that is typically much smaller. Thank you for pointing out the issue regarding overlapping assumptions.
>
> ### 5. Numerical evidence vs. Byzantine fraction
> In our current experiments, \delta = 0.2 does not directly represent the proportion of Byzantine clients among the sampled clients in each round; the number of Byzantine clients in the sampled set varies from round to round, and the behavior of baselines (e.g., truncated plots of FedAvg/FedAvgM) already reflects regimes where the sampled set may have a Byzantine majority. That said, we agree that explicit ablations over different Byzantine fractions would be much clearer.

---

### Official Review · Reviewer_VRBG · 2025-10-27

**Soundness:** 3
**Presentation:** 2
**Contribution:** 2
**Rating:** 2
**Confidence:** 4

**Summary:**

This work proposes D-Byz-SGDM algorithm for Byzantine-robust training, which can provably converge also in the partial participation regime, where only a subset of clients is selected. The main challenge is that in such a case, Byzantine clients can form a majority and destroy the training. The authors demonstrate that the proposed algorithm can converge even with partial participation by using momentum buffers from the previous round for the clients that were not selected in the current round. The authors provide convergence proof under the bounded gradient assumption and test the performance in training standard models against several baselines. Empirical results demonstrate the competitive performance of the proposed algorithm in practice.

**Strengths:**

- Detailed experimental setup, where all training details are properly described along with sweep details.

- The proposed method is tested in various settings: IID and Non-IID, under different attacks, models, and datasets. In all cases, the proposed algorithm achieves competitive performance. Surprisingly, it even outperforms FedAvg in the absence of Byzantine clients.

- The method is simple and easy to incorporate into existing code bases.

**Weaknesses:**

- Convergence under the bounded gradient assumption. This is the key limitation of this work in theory. The authors made a small note that it can be avoided. I encourage the authors to provide a discussion and a proof sketch, so that it can indeed be avoided. Otherwise, such claims do not bring any value.

- In the experiments, the authors compare against a modified version of Byz-VR-MARINA-PP (where I guess variance reduction is replaced by momentum averaging), instead of the one originally proposed. Therefore, the comparison against the original Byz-VR-MARINA-PP is missing, which is the only existing baseline in the literature. I encourage the authors to include a discussion on the modification of Byz-VR-MARINA-PP and explain why this modification was chosen over the original algorithm.

- Increased memory footprint as the server has to store all momentum buffers. This is not the case for Byz-VR-MARINA-PP, where only one vector of parameters should be stored. The authors are encouraged to provide a more detailed discussion on the algorithm design, its advantages, and disadvantages.

- The proof of lower bounds consist the restriction $p \ge \frac{-\delta + \sqrt{\delta^4+4}}{2}$, which must be specified in the statement. In my view, this is quite restrictive in opposite to what the authors claim. In particular, for $\delta=0.1$, it requires $p \gtrsim 0.95$, which means "almost" a full participation regime. Moreover, the choice made in experiments, $\delta=0.2$ and $p=0.5$, does not satisfy it. Finally, for any choice of $\delta$, we have $p \gtrsim 0.78$. I encourage the authors to recheck their proofs and provide a more relaxed condition on $p$ and $\delta$, which makes sense in real scenarios.

**Questions:**

- This sentence: "However, variance reduction methods perform poorly for deep learning model" is partially wrong and refers to old work. There are many recent successful variants of variance-reduced deep learning optimizers [1,2,3] that are competitive in practice.

[1] Yin, Yida, et al. "A coefficient makes svrg effective." arXiv preprint arXiv:2311.05589 (2023).

[2] Yuan, Huizhuo, et al. "Mars: Unleashing the power of variance reduction for training large models." arXiv preprint arXiv:2411.10438 (2024).

[3] Chang, Da, Yongxiang Liu, and Ganzhao Yuan. "On the Convergence of Muon and Beyond." arXiv preprint arXiv:2509.15816 (2025).

- How difficult is it to generalize the method to other sampling strategies (e.g., when we sample $S$ clients from all $n$ clients)?

- Some captions are confusing and not clear on the first read. The authors are encouraged to make all figures self-contained by explaining what changes from one line of figures to another (e.g., IID or Non-IID should also be specified)

- Why are FedAvg and FedAvgM not included in the Resnet-18 results?

- Can the authors provide more details behind (1)? In my view, this equality is not obvious and should be clearly derived. From the first glance, it looks like the RHS of (1) should scale with $\alpha$, not $\alpha^2$ (at least, it is the case in the full participation regime). Moreover, it is not obvious to me why the probability parameter $p$ is not involved in (1). I have similar questions regarding (2-6). I encourage the authors to provide more sophisticated derivations, where each step is clearly derived. Moreover, (1) contradicts the derivations in line 1524. Which one is actually correct?

- What does it mean "technically we can prove this but omit it for simplicity maybe icnluded to time 0" in lines 1438-1439? What to prove? How can the word "maybe" be used in such a claim?

- In line 1516, there is a typo; I believe it should be $\nabla f_i(x^{t-1})$, not $\nabla f(x^{t-1})$.

- Derivations in lines 1569-1588 are also unclear; the authors hide most of the steps and show only the final bound, which makes it difficult to check the correctness.

- In line 1397, the momentum parameter $\alpha$ satisfies $\alpha^2 \ge \frac{90L^2\eta^2}{p^2}$, while in the statement of theorem 13 $\alpha^2 = \min\{1, 9L\eta/p\}^2 \le 81L^2\eta^2/p^2$. Is this only a typo?

- In the statement of Theorem 13, why do the authors need such an initialization that decays with $n$, not $G$? Is it a typo?

---

> ### Author Response · Authors · 2025-12-03
> **Official Comment by Authors**
>
> We thank the reviewer for their careful reading and constructive feedback.
> ### 1. Regarding variance reduction methods in deep learning:
> We have revised the manuscript to clarify this distinction, and we agree that heavy-ball momentum can be viewed as a form of variance reduction.  In our paper, we used "variance reduction" in a narrower sense to refer specifically to SVRG-type methods (e.g., SVRG, SAGA) or large-minibatch schemes that explicitly reduce stochastic gradient variance by exploiting the finite-sum structure of the underlying problem. As noted by Defazio & Bottou (2019), such methods tend to perform poorly in deep learning settings. We do not criticize momentum-based approaches such as those in [2, 3].
>
> ### 2. On generalizing the method to partial client participation (e.g., sampling S clients from N):
> We view this as promising future work, though extending to arbitrary participation patterns remains an open direction. Our current proof technique relies on specific assumptions regarding client participation. As Reviewer TprE suggested, incorporating techniques from MIFA's analysis (which handles arbitrary participation) may help address this limitation. However, MIFA only considers vanilla SGD in a non-Byzantine setting, so non-trivial modifications would be required.
>
> ### 3. On figure captions:
> We have revised the figure captions to be more self-contained, now explicitly specifying the attack type, aggregation method, and data distribution (IID vs. non-IID) for each subplot.
>
> ### 4. On the absence of FedAvg and FedAvgM in the ResNet-18 results:
> FedAvg and FedAvgM are included in the ResNet-18 results. However, because these methods experience Byzantine majority in the first three rounds across the 200-epoch training, their curves collapse early and are difficult to discern visually. This is noted in the caption of Figure 1(b).
>
> ### 5. About proofs:
> Upon carefully revisiting these comments, we identified major errors in our upper-bound proofs that we were unable to resolve within the discussion period. We have removed the theoretical analysis section from the current revision and are temporarily withdrawing those results. We are committed to addressing these issues rigorously in future work.
>
> ### 6. On the initialization scaling with G rather than n in Theorem 13:
> Since we assume $G \geq n/2$, we can upper bound the expression accordingly. This choice was made to align our convergence bound with the formulation in Karimireddy et al. (2022).

---

### Official Review · Reviewer_thWo · 2025-10-31

**Soundness:** 3
**Presentation:** 3
**Contribution:** 2
**Rating:** 6
**Confidence:** 3

**Summary:**

This paper proposes distributed stochastic gradient methods with **asynchronous momentum updates** for **Byzantine-robust federated learning under partial client participation**, a setting where standard robust aggregation fails. The methods achieve **sub-linear convergence with residual error**, matching known lower bounds, and are validated empirically on MNIST and CIFAR-10, showing superior accuracy across various robust aggregators and data heterogeneity levels. Key limitations include reliance on restrictive assumptions, lack of analysis for communication delays, and missing discussion of related work on momentum-based Byzantine-robust methods.

**Strengths:**

- This paper proposes distributed stochastic gradient methods with asynchronous momentum updates under **partial client participation** for solving Byzantine optimization problems. It is the first Byzantine-robust method capable of handling partial client participation, a crucial feature for federated learning (FL) settings.  The main technical difficulty is that the robust aggregation strategy under partial client participation fails to be Byzantine-robust. The proposed solution is to leverage asynchronous/delayed momentum aggregation protocols.
- The proposed methods achieve robustness against Byzantine clients while attaining **sub-linear convergence with residual error**, matching known lower bounds. This indicates the tightness of the derived convergence guarantees.
- **Extensive experiments**, including training a convolutional network on MNIST and a ResNet-18 on CIFAR-10, demonstrate that the proposed algorithms consistently achieve superior convergence performance in terms of final accuracy. These results hold across various settings, including different robust aggregators and levels of data heterogeneity, in both Byzantine and non-Byzantine scenarios.

**Weaknesses:**

- The convergence results are limited because they rely on three restrictive assumptions: Assumptions 3, 4, and 5. First, Assumptions 3 and 5 appear redundant. These assumptions were not required in Gorbunov et al. (2023), so removing one could simplify the analysis without affecting convergence. Second, Assumption 4 is stronger than the $(G, B)$-gradient dissimilarity condition used in [1], which may unnecessarily restrict the applicability of the results.



- The analysis does not impose any assumptions on communication delays in the asynchronous protocol. Consequently, Theorem 6 does not capture the effect of asynchrony on the stepsize or convergence of the proposed algorithms. This is unusual, as asynchronous or delayed gradient-based methods, e.g., [2,3], typically require the stepsize to account for the delay, which directly impacts the convergence bounds.

Ref:

[1] Allouah, Y., Guerraoui, R., Gupta, N., Jellouli, A., Rizk, G., & Stephan, J. (2024). Adaptive gradient clipping for robust federated learning. _arXiv preprint arXiv:2405.14432_.

[2] Feyzmahdavian, H. R., & Johansson, M. (2023). Asynchronous iterations in optimization: New sequence results and sharper algorithmic guarantees. _Journal of Machine Learning Research, 24_(158), 1–75.

[3] Gurbuzbalaban, M., Ozdaglar, A., & Parrilo, P. A. (2017). On the convergence rate of incremental aggregated gradient algorithms. _SIAM Journal on Optimization, 27_(2), 1035–1048.

**Questions:**

- One prior work closely related to this paper is [1], as it also proposes algorithms using momentum updates in Byzantine settings. It would strengthen the manuscript to include a discussion of [1] and clarify how the proposed methods compare or differ from it.

[1] Özfatura, K., Özfatura, E., Küpçü, A., & Gunduz, D. (2023). Byzantines can also learn from history: Fall of centered clipping in federated learning. _IEEE Transactions on Information Forensics and Security, 19_, 2010–2022.

---

> ### Author Response · Authors · 2025-12-03
> **Official Comment by Authors**
>
> We thank the reviewer for their careful reading and constructive feedback. We address each concern below.
> ### 1. Regarding the assumptions (Assumptions 3, 4, and 5):
> We would like to emphasize that the bounded variance assumption we adopt is widely accepted in the stochastic optimization literature and does not significantly limit the practical applicability of our results.
> We acknowledge that our assumptions are indeed stronger than the (G,B)-gradient dissimilarity condition used in (Allouah et al., 2024). However, we note that Allouah et al. (2024) does not provide convergence bounds for the stochastic setting, which is the focus of our work. Additionally, while the local Hessian variance assumption is weaker and commonly used in SVRG-type optimizers, it is not a natural assumption to our method since we do not employ minibatch difference estimators. The bounded variance assumption we adopt is widely accepted in the stochastic optimization literature and does not significantly limit the practical applicability of our results.
>
> ### 2. Regarding the effect of asynchrony on the stepsize and convergence:
> Our design choice allows us to avoid explicit delay-dependent stepsize conditions while still accounting for the irregularity in updates.  We would like to clarify that our method is not truly asynchronous in the traditional sense. Instead, the effect of communication patterns in our protocol is captured through the Bernoulli parameter p, which directly influences the convergence behavior.
>
> ### 3. Regarding the comparison with [1]:
> We note that [1] primarily proposes an attack method and does not provide convergence guarantees, which makes a direct comparison of convergence results infeasible.  We appreciate the suggestion to discuss [1] in more detail. However, since our focus is on developing algorithms with provable convergence bounds under Byzantine settings, we added a clarifying remark in the related work section to highlight this distinction.

---

### Meta-Review · Area_Chair_CWxr · 2025-12-30

**Summary:**

This paper proposes a new method for federated learning with Byzantine workers under partial participation. The main idea for tolerating a potentially majority of malicious clients is to consider all clients, including those that do not participate, by incorporating their delayed updates. However, during the review process, two reviewers identified critical issues in the theoretical analysis. These concerns were later confirmed by the authors, who subsequently removed the theoretical results from the submission. While the experimental results are promising and the method itself is novel, either a more thorough empirical evaluation on larger, real-world applications is required, or the theoretical guarantees need to be corrected and reinstated.

Therefore, I recommend rejecting the paper at this time. That said, I believe that a future submission addressing these issues would make a valuable contribution to the collaborative learning community.

**Reviewer Concerns:**

The main concerns are related to the original theoretical results, as outlined in the summary above.

**Reviewer Scores:**

Based on the identified issues with the theory, I believe all the reviewers would either keep or decrease their current score.

---

### Decision · Program_Chairs · 2026-01-26

Reject